# Targeting Mast Cells in Allergic Disease: Current Therapies and Drug Repurposing

**DOI:** 10.3390/cells11193031

**Published:** 2022-09-27

**Authors:** Jason R. Burchett, Jordan M. Dailey, Sydney A. Kee, Destiny T. Pryor, Aditya Kotha, Roma A. Kankaria, David B. Straus, John J. Ryan

**Affiliations:** Department of Biology, Virginia Commonwealth University, Richmond, VA 23284, USA

**Keywords:** allergy, inflammation, mast cell, IgE, atopy, repurposing, repositioning, statin, JAK, STAT

## Abstract

The incidence of allergic disease has grown tremendously in the past three generations. While current treatments are effective for some, there is considerable unmet need. Mast cells are critical effectors of allergic inflammation. Their secreted mediators and the receptors for these mediators have long been the target of allergy therapy. Recent drugs have moved a step earlier in mast cell activation, blocking IgE, IL-4, and IL-13 interactions with their receptors. In this review, we summarize the latest therapies targeting mast cells as well as new drugs in clinical trials. In addition, we offer support for repurposing FDA-approved drugs to target mast cells in new ways. With a multitude of highly selective drugs available for cancer, autoimmunity, and metabolic disorders, drug repurposing offers optimism for the future of allergy therapy.

## 1. Introduction

The burden of allergic disease and asthma has been increasing in recent decades with an estimated 20% of the worldwide population affected and wide variations between countries [1]. With rising prevalence comes increased stress on healthcare systems and economic burden due to treatment. In 2013, the total cost of asthma alone in the United States was USD 81.9 billion [2]. Current therapeutics have inadequately controlled the growing epidemic of atopic disease; discovery of improved and cost-effective treatments is necessary.

There are several treatments approved for controlling atopic disease, all of which have limitations. One of the most common treatments for asthma are inhaled corticosteroids (ICS). ICS, such as budesonide, are effective at controlling new onset asthma but if control diminishes, doses may have to increase to quadruple the original dose [3]. Antihistamines, another mainstay treatment for atopic disease, are widely available over the counter and work by antagonizing the H-1 or H-2 receptors. There is also a growing body of publications regarding antagonizing the H-4 receptor [4]. Other treatments for asthma and allergy include long-acting β-agonists (LABA), leukotriene antagonists, targeted biologics, and allergen specific immunotherapy [5].

The discovery, development, and approval process of new therapeutics can require 15 years and an estimated USD 2B. It also suffers from high attrition rates [6]. This large investment in time and money, with an approximate 45% failure rate, makes drug repurposing an attractive option for researchers and pharmaceutic companies [7]. Drug repurposing (also referred to as drug repositioning, redirecting, reprofiling, etc.) is the relatively recent concept of employing drugs outside their original clinical use. This can mean finding new diseases in which the intended drug target is useful or leveraging off-target drug effects for new clinical use [7]. Drug development via repurposing is far faster, approximately 80% cheaper, and has a reduced failure rate [8]. A recent high-profile example is the rapid repurposing of multiple drugs to address the COVID-19 pandemic [9]. Key advantages of drug repurposed drugs include established safety profiles and known molecular targets. If these targets are fundamental biological mechanisms, they are often shared across cell types. 

Mast cells are major contributors to allergic disease, producing mediators such as cytokines, leukotrienes, and histamine that have well established roles in atopic pathology. Mast cell hyperplasia is documented in the asthmatic lung, as are elevated levels of tryptase, chymase, and histamine in sputum and lung lavage [10,11,12,13]. Evidence for mast cell involvement in allergic disease is also documented using mast cell-deficient mouse models. For example, systemic or cutaneous anaphylaxis induced by passive IgE sensitization is completely absent in mice lacking mast cells [14,15]. More complex models also show a major role for mast cells. For example, peanut antigen-induced anaphylaxis is greatly reduced in mast cell-deficient mice [16]. Similarly, mouse models of allergic airway hyperresponsiveness (AHR) demonstrate a critical role for mast cells, particularly their airway infiltration, and agree with data suggesting mast cells increase AHR in asthmatics [17,18,19,20,21,22]. In addition to true allergic responses, pseudo-allergic responses induced by drugs, bacterial products, and neuropeptides employ the MAS-related G-protein coupled receptors (MRGPR) expressed by mast cells and correlate with mast cell activation [23]. Therefore, specifically targeting mast cell function is a logical goal for treating allergic disease. Current drugs targeting mast cells are limited and have failed to contain their growing rate of allergic disease. As mast cells employ fundamental molecular mechanisms already well studied, they are a logical candidate for the growing trend of drug repurposing. This review will summarize current treatments targeting mast cells and explore potential drugs with potential for repurposing in allergic disease. 

## 2. Targeting Mast Cells with the Latest Drugs

### 2.1. Monoclonal Antibody-Based Drugs 

Antibodies are an attractive therapeutic option given their high affinity and specificity of target binding. Monoclonal antibodies, originally produced from B cell hybridomas, have a single specificity and provide a reliable, reproducible drug which can be modified through genetic manipulation to optimize activity [24,25,26,27,28]. Some monoclonals are already in use to treat allergic disease and there is the potential to repurpose others initially developed to fight other diseases. 

#### 2.1.1. Omalizumab and Ligelizumab Antagonize IgE-FcεRI Binding 

Omalizumab, commercially known as Xolair, is a monoclonal antibody approved by the U.S. Food and Drug Administration (FDA) and the European Union as a therapeutic for treating IgE-induced asthma and chronic spontaneous urticaria [29]. A notable study emphasized that omalizumab binds to Cε3 with picomolar affinity [29]. Cε3 is the third constant domain of the heavy chain of circulating IgE, and critical for IgE binding site to the α2 subunit of the FcεRI and FcεRII (CD23) receptors [29]. FcεRI is the high-affinity IgE receptor, a tetrameric complex consisting of an IgE-binding alpha subunit, and a beta subunit that induces cell signaling in conjunction with a dimer of gamma subunits. FcεRII is the low-affinity IgE receptor. It is important to note that omalizumab does not activate mast cells [29]. Omalizumab develops immune complexes with free IgE in serum, sterically hindering IgE interactions with FcεRI and FcεRII receptors [29,30]. Omalizumab reduces free IgE serum levels, which results in FcεRI downregulation on basophils and mast cells [31,32,33]. One study found that the therapeutic effects of omalizumab are a result of eliminating an activation signal rather than producing an inhibitory signal in mast cells [31]. 

Ligelizumab is a monoclonal antibody currently in phase 3 clinical trials for chronic spontaneous urticaria. This monoclonal antibody is also an anti-IgE therapeutic and a possible successor to omalizumab [34]. Ligelizumab has high binding affinity for IgE. A 1:1 molar ratio of ligelizumab to IgE inhibited 90% of IgE-mediated human mast cell degranulation [35]. Ligelizumab suppressed free IgE serum levels and FcεRI receptor expression with a dose-dependent and time-dependent manner [36]. While ligelizumab’s central mechanism of action is neutralizing free serum IgE levels, it was observed to remove IgE-Fc3-4 fragments from the FcεRIα receptor [34]. This is a significant advance, since IgE is bound with high affinity and a long half-life that allows FcεRI accumulation [35]. Thus, displacing IgE from FcεRI could lead to a faster and more efficacious treatment.

#### 2.1.2. Dupilumab Suppresses IL-4 and IL-13 Signaling

IL-4 and IL-13 are more important for eliciting allergy than perhaps any other soluble factors. These two cytokines are produced by mast cells and Th2 cells and have overwhelming influence on foundational disease aspects, including Th2 differentiation, B cell isotype switching to IgE, mast cell proliferation and survival, goblet cell metaplasia, and mucus production. These cytokine functions have been reviewed in several recent articles [37,38,39,40,41]. For this reason, there has long been interest in suppressing IL-4 and IL-13. This culminated in the recent deployment of dupilumab, sold as Dupixent, a monoclonal antibody inhibiting IL-4 and IL-13 signaling [42]. This drug has received approval in the United States and the European Union to treat allergic disorders, including atopic dermatitis and allergic asthma [42]. Dupilumab binds to the IL-4Rα subunit with high affinity [43]. IL-4 signals by inducing complex formation between IL-4Rα and the common gamma (γc) subunits, while IL-13 employs complexes of IL-4Rα and IL-13Rα1 [43]. By effectively blocking both IL-4 and IL-13 function, dupilumab suppresses the two major signals driving Type 2 immune responses. An additional study mentioned that dupilumab downregulates genes that are in the Th2 immune response pathway [42]. Dupilumab administered for 16 weeks decreased IL-4 and/or IL-13 regulated Th2 response gene expression in skin and serum samples of atopic dermatitis patient*s* [42]. These downregulated genes include CCL17, CCL18, CCL26, and DPP4 [42]. This study also found that dupilumab inhibited mRNA expression of the inflammatory proteases MMP12 and SERPINB4 and the surface receptors ICOS and CCR7 involved in T-cell activation [42]. Dupilumab was also found to suppress genes including IL17A, IL22, and S100As in the Th17 and Th22 pathways [42]. Downregulation of these genes by dupilumab reduced the probability of mast cell activation [42]. 

In addition to dupilumab, a recent paper raises the exciting possibility of neutralizing IL-4 and IL-13 through vaccination. Conde and co-workers showed that immunizing mice with IL-4 or IL-13 conjugated to a mutant, nontoxic diphtheria toxin elicited persistent neutralizing antibodies that greatly diminished asthma-like symptoms in a mouse model of allergic airway disease [44]. This included decreased pulmonary eosinophilia, mucus production, and airway hyperresponsiveness. Moreover, the strategy was also effective in a mouse engineered to express human IL-4, IL-13 and IL-4Rα. The obvious benefit of vaccination is decreased cost. While safety concerns and human testing need considerable investigation, this represents a novel approach to allergy therapy.

#### 2.1.3. Rituximab Reduces IgE Levels by Depleting B Cells

Rituximab, also known as Rituxan, is a monoclonal antibody that is used as a therapeutic for autoimmune diseases, such as rheumatoid arthritis and for B cell malignancies [45]. Rituximab targets the CD20 protein that is located on B cells, which allows rituximab to deplete B cells [46]. Despite this intended target, there is evidence that rituximab suppresses mast cell responses. A study analyzing 15 patients with primary Sjögren’s syndrome found that one year rituximab treatment led to a 50% depletion of mast cells in the salivary glands with no associated decrease in T cells [46]. Furthermore, rituximab promoted mast cell apoptosis in vitro when mixed with heterologous lymphocytes. One explanation put forth by the authors for this unexpected finding is that rituximab may interact with the FcεRI β subunit, which is structurally homologous to CD20 [46]. 

An obvious potential use for rituximab in allergic disease is depleting IgE-secreting B cells. This drastic approach could be justified for severe diseases such as poorly controlled asthma. The efficacy of this approach is unclear. A 2013 study found no change in serum IgE after 6 months of rituximab therapy among 47 patients [47]. However, a recent retrospective study of patients treated with rituximab for eosinophilic granulomatosis with polyangiitis (EGPA) found that among a subset with asthma, 12 of 17 had disease remission [48]. A similar work found that while rituximab suppressed asthma in EGPA patients, these symptoms recurred in many patients after a 2-year follow up [49]. Collectively, these publications point to the possibility of suppressing severe allergic disease with rituximab. The efficacy of this treatment and its possible off-target killing of mast cells warrants further study.

#### 2.1.4. Lirentelimab Activates Siglec-8

Lirentelimab is a monoclonal antibody that is currently in phase 3 clinical trials for eosinophilic gastritis and duodenitis. This monoclonal antibody, also known as AK002, has potential for becoming a therapy that depletes eosinophils and inhibits mast cell function [50]. Lirentelimab targets sialic acid-binding immunoglobulin-like lectin 8 (Siglec-8), which is primarily expressed on mast cells and eosinophils [50]. Siglec-8 negatively regulates tyrosine-based activation motif receptors, such as those on FcεRI [51]. Lirentelimab appears to elicit eosinophil depletion via NK cell-mediated killing [52]. It also acts by eliciting Siglec-8-mediated inhibitory signals [52]. This leads to a significant decrease in eosinophils and mast cells in the inflamed tissues and improved symptom burden [53].

Since lirentelimab engages Siglec-8, it may promote the recruitment of phosphotyrosine phosphatases SHP-1 and SHP-2 and the inositol phosphatase SHIP-1, inhibitory FcεRI signals [54]. In agreement with this, the antibody suppressed FcεRI-dependent Ca^2+^ flux, β-hexosaminidase and histamine release, and cytokine and chemokine secretion in cultured human mast cells [51,55]. Similar inhibition was observed in vitro with human lung tissues, as was a reduction in FcεRI-mediated contraction of human airway bronchial rings [51]. Therefore, lirentelimab appears to act both by cell depletion and Siglec-8-mediated inhibition of FcεRI signaling [51]. 

#### 2.1.5. MTPS9579A Disrupts Tryptase Tetramers

MTPS9579A is a novel monoclonal antibody currently in phase 2 clinical trials for allergic asthma. This full-length humanized IgG4 antibody binds human tryptase with high affinity [56]. Tryptase is a serine protease, and a major component of secretory granules in human mast cells [56,57]. Tryptase is only active in its tetrameric form, which is released during mast cell degranulation [56,58]. Active tryptase secreted into extracellular space induces airway hyper-responsiveness, bronchoconstriction, and heightened mast cell degranulation in allergic asthma [56,59]. Elevated tryptase is found in bronchoalveolar lavage fluid and correlates with increased disease severity [56,60]. 

MTPS9579A acts at least partly by dissociating tetrameric tryptase into inactive monomers [56]. In human trials, the mAb suppressed tetrameric tryptase while increasing total tryptase in the nasal mucosa [61]. One study showed that MTPS9579A diminished IgE-mediated systemic anaphylaxis in a humanized mouse model and reduced tetrameric tryptase levels in the lung tissue of allergen-challenged cynomolgus monkeys [56]. These studies suggest that MTPS9579A may be a viable therapeutic for allergic asthma by disrupting tryptase function [56]. 

#### 2.1.6. CDX-0159 Targets KIT Receptor

CDX-0159 is a novel monoclonal antibody in phase 1 clinical trials for chronic spontaneous urticaria and chronic inducible urticaria (NCT04538794) and phase 2 for chronic spontaneous urticaria (NCT05368285). This mAb targets c-kit/CD117 receptor, a critical stimulus for mast cell migration, adhesion, and activation [62]. CDX-0159 was derived from mice immunized with the membrane proximal domain of human KIT [62]. The mAb reduces SCF binding and KIT tyrosine phosphorylation, apparently acting as an allosteric inhibitor. The result is a near absence of serum tryptase levels in human subjects, suggesting mast cell depletion, and a 4-fold increase in plasma SCF likely due to KIT blockade. Importantly, CDX-0159 contains 3 mutations in its Fc domain that preclude FcγR interactions and subsequent inflammatory reactions. Further, the mutations increase affinity for neonatal Fc receptor FcRn, which promotes recycling after cellular uptake and extended the serum half-life to 32 days [62].

Overall, monoclonal antibodies currently available or in clinical trials could treat mast cell disorders by targeting multiple aspects of mast cell function. They demonstrate the myriad ways mAb can be engineered to selectively antagonize protein targets with few side effects and the ability to modify the IgG structure to avoid unwanted signals such as FcγR activation. This is a significant growth area for treating allergic disease. For a summary of the mAb discussed in this section, see Table 1.

### 2.2. DARPins

Designed ankyrin repeat proteins (DARPins) are an exciting, perhaps field-changing approach to disease treatment. These approximately 14–21 kDa proteins can be screened for effector functions including receptor agonist or antagonist activity much like mAbs, but they can be produced in *E. coli* and have excellent pharmacokinetics [63]. Allergy-related DARPin use is an emerging area showing promise. Eggel et al. demonstrated their potential utility in a paper showing DARPins could dissociate IgE from FcεRI on human basophils and suppress passive cutaneous anaphylaxis [64]. A follow-up study described another DARPin capable of inhibiting FcεRI-IgE interactions while also engaging the suppressive FcγRII (CD32) receptor [65]. Another study showed that a DARPin interacting with the low affinity IgE receptor on B cells, CD23, suppressed IgE production [66]. Thus, while DARPins are not yet well documented as potential allergy therapeutics, it seems likely that progress may come quickly.

### 2.3. Kinase Inhibitors 

The IgE-induced release of mast cell mediators like histamine, cytokines, and proteases is critical to causing inflammatory reactions observed in allergic diseases. The responses that follow aggregation of IgE-occupied FcεRI receptors are enhanced by SCF-KIT signaling. FcεRI and KIT employ an array of tyrosine and lipid kinases [67]. As a result, drugs that inhibit kinase activity can be repurposed to treat mast cell activation and allergic diseases. The drugs discussed in this section are summarized in Table 2.

#### 2.3.1. BCR-ABL Tyrosine Kinase Inhibitors

The mutant BCR-ABL tyrosine kinase plays an important role in malignancy and has been the subject of an intensive research for inhibitors. Several inhibitors targeting BCR-ABL have been identified, which vary in efficacy and mechanism. First-generation inhibitors include imatinib; second-generation inhibitors include nilotinib, dasatinib, and bosutinib; and third-generation inhibitors include ponatinib. While originally developed for BCR-ABL-expressing neoplasias like chronic myelogenous leukemia (CML), these drugs have off-target effects on kinases important in mast cell disorders, including KIT, Src and Btk, [68,69]. For example, El-Agamy et al. demonstrated that nilotinib inhibited anaphylaxis induced by either compound 48/80 or IgE-antigen in rodents, and reduced histamine and TNF release from antigen-treated rat peritoneal mast cells [70]. In a study by Cahill et al., patients who had severe refractory asthma and airway hyperresponsiveness were treated with imatinib [71]. After six months, imatinib decreased mast cell counts, serum tryptase (a measure of mast cell activation), and airway hyperresponsiveness [71]. The potent effects of BCR-ABL inhibitors may be most evident in patients on long-term (> 24 months) imatinib therapy, who show a 90% reduction in mast cells and serum tryptase [72]. These studies support further attempts to repurpose BCR-ABL inhibitors for mast cell-mediated diseases.

#### 2.3.2. Bruton’s Tyrosine Kinase Inhibitors

Another relevant class of inhibitors target Bruton’s tyrosine kinase (BTK). BTK plays an important role in transducing FcεRI crosslinking signals into mediator release from basophils and mast cells [73]. Beyond allergic disease, BTK has many roles, especially in B cell proliferation and malignancy [74]. For this reason, BTK has been the focus of attempts to suppress B cell proliferation. Ibrutinib (PCI-32765) is a first-generation BTK inhibitor, and acalabrutinib is a second-generation drug in this family. Both are FDA-approved, irreversible inhibitors that are administered orally [75].

Ibrutinib is approved for the treatment of B cell malignancies [36]. BTK is required for B cell activation after the B cell antigen receptor has been engaged [76]. Since BTK is critical for B cell signaling and survival, ibrutinib has been effective, even with chronic use, in patients with B cell malignancies [36]. However, a study by Chang et al. showed that ibrutinib not only targets B cells but also macrophages, monocytes, and mast cells [77]. In the study, ibrutinib was administered in four different mouse disease models: passive cutaneous anaphylaxis (PCA), reversed passive anaphylactic reaction (RPA), collagen-induced arthritis (CIA), and collagen antibody-induced arthritis (CAIA) [77]. Ibrutinib reversed arthritic inflammation in a dose-dependent manner in the CIA model and blocked clinical arthritis from developing in the CAIA model [77]. In the CIA and CAIA models, the BTK inhibitor also inhibited macrophages and monocytes from infiltrating the synovium and preserved joint cartilage and bone integrity [77]. In the PCA model, ibrutinib reduced inflammation [77]. Further, it decreased production of TNF-α, IL-1β, and IL-6 in primary monocytes after FcγR stimulation and inhibited the release of histamine, PGD2, TNF-α, IL-8, and MCP-1 in human mast cells after FcεRI stimulation [77]. As ibrutinib was effective in disease models that did not depend on B cell antibody production, Chang et al. found that the BTK inhibitor targeted not only B lymphocytes but also mast cells and monocytes, which are critical BTK-expressing effector cells in arthritis and allergy [77]. Other studies suggest that ibrutinib can suppress mast cell proliferation but likely does not induce significant mast cell apoptosis. Smilijkovic et al. showed ibrutinib inhibited HMC-1 mast cell line proliferation at concentrations above 1 μM, compared to an IC_50_ of 30 nM to inhibit IgE-mediated basophil degranulation [78]. Gamperl et al. similarly noted that ibrutinib suppressed proliferation of canine mastocytoma lines with an IC_50_ of 0.5–3 μM, but apoptosis required 10–25μM drug concentrations [79]. Since ibrutinib plasma concentrations are approximately 0.5 μM, its effects on BTK-mediated mast cell activation may be more important that its anti-proliferative or apoptosis-inducing abilities [36].

Subsequent studies have expanded upon the findings of Chang et al. For instance, Dubovsky et al. confirmed that ibrutinib is not only an irreversible inhibitor of BTK but also a reversible inhibitor of interleukin-2-inducible kinase (ITK), which plays an important role in T cell signaling [80]. The authors found that ibrutinib inhibits the development of Th2, but not Th1, immunity [80]. This selective role in Th2 differentiation can have important therapeutic applications [80]. Furthermore, Dispenza et al. studied the effects of both ibrutinib and acalabrutinib on human mast cells [75]. In the study, the inhibitors prevented IgE-mediated cytokine production and degranulation in human basophils and primary mast cells [75]. They also blocked IgE-mediated bronchoconstriction in human lung tissue [75]. Additionally, acalabrutinib completely prevented moderate IgE-mediated passive systemic anaphylaxis (PSA) and reduced death during severe PSA [75]. The findings suggest that BTK inhibitors could be beneficial in human anaphylaxis [75].

Studies have been conducted on the efficacy of ibrutinib therapy on humans with a history of allergic disease. One investigation conducted by Regan et al. found that ibrutinib eliminated reactivity to the basophil activation test and the aeroallergen skin test in patients with a history of allergy [81]. Another found that, in adults with tree nut or peanut allergy, short-term ibrutinib treatment (as few as two doses) eliminated IgE-mediated basophil activation and attenuated responses to skin prick tests [82]. The short-term therapy was well-tolerated in patients and significantly reduced or eliminated patients’ basophil and mast cell reactivity to food allergens [82]. These findings have important implications both for allergy therapy and for interpreting allergy testing of patients treated with BTK inhibitors.

#### 2.3.3. Lyn, Fyn, and Syk Tyrosine Kinase Inhibitors

Lyn, Fyn, and Syk tyrosine kinases mediate the earliest FcεRI signaling events. Lyn is pre-associated with the FcεRI beta chain and phosphorylates the immunoreceptor tyrosine-based motifs (ITAMs) of FcεRI β and γ subunits. Recruitment to these phosphorylated ITAMs activates cytoplasmic Syk, which phosphorylates phospholipase C, gamma 1 (PLCγ1), and Linker for activation of T cells (LAT). Fyn is also rapidly activated and stimulates a pathway employing GRB2-associated-binding protein 2 (GAB2)/phosphatidylinositol 3 kinase (PI3K). While these pathways are nuanced and have overlapping roles, FcεRI can be viewed as having two key signaling pathways: Lyn/Syk/LAT and Fyn/GAB2/PI3K [83,84]. Therefore, Lyn and Fyn serve apical roles that make them logical drug targets in allergic disease. At present, there are no selective drugs targeting these kinases, but several drugs have off-target inhibitory activity that is relevant to these pathways.

Park et al. studied the anti-inflammatory effects of AZD7762 (3-[(aminocarbonyl)amino]-5-(3-fluorophenyl)-N-(3S)-3-piperidinyl-2-thiophenecarboxamide) [85]. AZD7762 is a checkpoint kinase inhibitor that also inhibits Lyn, Fyn and Syk and is being tested as an anti-cancer drug [85]. AZD7762 inhibited antigen-induced degranulation and cytokine production in bone marrow-derived mast cells (BMMCs), degranulation in human mast cells, and PCA in mice [85]. The drug suppressed IgE-mediated mast cell activation and allergic responses, which suggests that it could be used to treat allergic diseases mediated by mast cells [85]. Another cancer drug, WZ3146 (N-[3-[5-chloro-2-[4-(4-methylpiperazin-1-yl)anilino]pyrimidin-4-yl]oxyphenyl]prop-2-enamide) that targets the epidermal growth factor receptor tyrosine kinase, also holds promise in allergy. Park et al. found that WZ3146, like AZD7762, reduced IgE-mediated Lyn, Fyn, and Syk activity as well as degranulation in BMMCs and human mast cells and IgE-dependent PCA in mice [86]. Finally, the BCR-ABL inhibitor Dasatinib also antagonizes Lyn, Fyn, and Syk activity and reduced antigen-induced BMMC degranulation and cytokine, and inhibited PCA in mice [87]. These studies support the idea that off-target effects of tyrosine kinase-selective inhibitors may prove beneficial in allergic disease.

#### 2.3.4. PI3K/AKT/mTOR Pathway Inhibition

PI3K promotes activation of the Akt kinase and mammalian target of rapamycin (mTOR), collectively termed the PI3K/AKT/mTOR pathway. The functional role of this cascade includes promoting cellular metabolism and growth, increasing protein translation, and inhibiting apoptosis [88]. The role of PI3K downstream of KIT and FcεRI signaling has been discussed in several review articles [89,90]. Being a vital pathway for a multitude of cellular functions, manipulating this pathway could have therapeutic effects and unintended consequences. There are numerous studies investigating the role of PI3K and its downstream partners in human disease; this review will discuss some potential options for therapeutics within the context of MC function. 

PI3K consists of a regulatory subunit most often referred to as p85, although mammals possess three genes encoding 5 regulatory subunits ranging from 50–85 kDa. The regulatory subunit forms heterodimeric complexes with p110 catalytic subunits p110α, p110β, p110γ, or p110δ [91]. There have been extensive efforts to generate PI3K inhibitors, leading to currently approved drugs [91]. Many of these drugs are designed to antagonize tumor growth, but they hold promise in mast cell-associated disorders. For example, skin-derived mast cells employ PI3K downstream of both FcεRI and the Mas-related G protein-coupled receptor member X2 (MRGPRX2) to elicit degranulation [92]. PI3K inhibition with the FDA-approved drug pictilisib essentially ablated granule release by skin mast cells, with a much more potent effect than MAPK inhibition [92].

PI3K generates PIP3, a membrane lipid that serves as a site of binding and activation for Akt and kinases in the TEC family [93]. While many TEC kinases have important roles, Akt has been the focal point of most studies [94]. Otherwise known as protein kinase B (PKB), Akt consists of three mammalian isoforms, Akt 1, 2, and 3 that play complex roles reviewed previously [95]. AKT signaling is fundamental to an array of cellular functions including growth, proliferation, metabolism, survival, and apoptosis [95,96,97,98]. Aberrant Akt signaling has been implicated in numerous diseases. In addition to cancer, where its oncogenic effects are established [99], AKT has been implicated in diabetes, cardiovascular disease, neurological disease, and allergic asthma [99,100]. AKT function specific to MCs includes promoting cytokine production via the transcription factors nuclear factor kappa B (NF-κB), nuclear factor of activated T cells (NFAT), and activator protein 1 (AP-1) [101]. Thus, Akt targeting could prove beneficial.

AKT has a critical role in drug resistant breast cancer [102], prompting drug development that might be leveraged in allergic disease. MK2206, a selective allosteric Akt inhibitor, completed phase 2 clinical trials for breast cancer treatment, where it showed antitumor effects [103,104]. In experiments using mice, MK2206 treatment reduced Akt activation and correlated with inhibited airway inflammation and diminished levels of the damage-associated molecular pattern (DAMP) high-mobility group box-1 (HMGB1) [105]. HMGB1 induces degranulation and inflammatory cytokine release via the RAGE/NF-κB pathway [106]. Mast cell migration induced by histamine—H4 receptor signaling was also attenuated by MK2206 [107]. Thus MK2206, and Akt targeting in general, shows potential as a means of antagonizing mast cell migration and function. 

Like other proteins in the PI3K/AKT/mTOR pathway, mTOR plays key roles in cell growth, proliferation, survival, and metabolism [108]. It has been extensively studied and reviewed in the context of cancer, metabolic disease, diabetes, and age-related diseases [109,110]. mTOR contributes to these myriad functions by participating in two distinct protein complexes, mTORC1 and mTORC2. Rapamycin, an extensively studied mTORC1 inhibitor, reduced inflammation in an IgE-mediated food allergy mouse model and inhibited IgE-induced mast cell cytokine production without affecting degranulation [111]. Other mTOR inhibitors have also shown promise in potentially inhibiting mast cells. In human mast cells stimulated by neuropeptides, the expression and release of pro-inflammatory mediators was inhibited by the flavonoid-based mTORC1 inhibitor 3′,4′,5,7-tetramethoxyflavone (methoxyluteolin) [112]. 

While these data indicate a pro-inflammatory role for mTOR, other studies showed the paradoxical finding that mTOR hyperactivation also suppresses IgE-mediated mast cell function. The mTOR activator MHY1485 is an mTOR agonist that binds mTORC1, amplifies its signal, and causes an inhibitory effect on the mTORC2 pathway [113]. A different mTOR activator, 3-benzyl-5-((2-nitrophenoxy) methyl)-dihydrofuran-2(3H)-one (3BDO), yielded similar results in a separate study [114]. Both mTOR agonists inhibited FcεRI-mediated degranulation and suppressed IL-6 and TNFα production at the transcriptional level in mouse mast cells. This coincided with decreased AKT phosphorylation by the AKT/mTORC2 complex [113,114]. An AKT agonist was shown to overcome the inhibitory effects, supporting the conclusion that mTORC2-mediated Akt activation is critical for mast cell function and can be targeted with small molecule approaches targeting mTOR [114]. 

The potential utility of rapamycin supports the broader concept of targeting cellular metabolism, specifically glycolysis. The PI3K/Akt/mTOR pathway promotes glycolysis as well as amino acid uptake and protein synthesis, processes critical for the energy demands of inflammation [115]. Our understanding of immunometabolism is growing rapidly and should reveal more targets and useful inhibitors that prove useful for suppressing mast cell function and allergy.

### 2.4. Targeting Critical Transcription Factors 

Transcription factors control a complex cytokine and growth factor network that determines mast cell differentiation, migration, and activation [116]. Therefore, transcription factors are a logical candidate for therapeutic development. We will discuss advances in targeting specific transcription factors with known roles in mast cell function and allergic disease. The drugs discussed in this section are summarized in Table 3.

#### 2.4.1. Inhibiting NFκB

FcεRI signaling elicits NFκB activation by inducing degradation of the protein inhibitor of κB (IκB), with subsequent NFκB nuclear translocation [117]. NFκB elicits expression of proinflammatory cytokines including TNFα and IL-6 [118]. Therefore, inhibiting NFκB would interfere with inflammatory mediator production and should attenuate the allergic response. The canonical NFκB cascade includes multiple molecules that can be targeted to disrupt cytokine transcription. Drugs with these effects are being developed for cancer and inflammatory diseases and might be repurposed for allergy. 

A 2010 study found that 19 of 2800 FDA-approved drugs screened could inhibit NFκB transcriptional function [119]. Other drugs such as bortezomib act indirectly by targeting the proteasome complex [120]. Most of these drugs have not been tested for effects on mast cell function and warrant investigation. Bortezomib inhibits calcium ionophore- or PGE2-induced human mast cell degranulation and cytokine secretion and induces apoptosis in mast cell leukemia, though this is attributed to the unfolded protein response [120,121]. Similarly, vorinostat (also known as SAHA) also induces cell death in the transformed human mast cell line HMC-1.2. However, it is unclear if this is due to the drug’s actions as a histone deacetylase inhibitor or the subsequent inhibition of NFκB [122]. Finally, it is worth noting that newer inhibitors support the premise of NFκB targeting in mast cell disorders. Matsumoto and colleagues developed a novel NFκB inhibitor, dehydroxymethylepoxyquinomicin (DHMQ), that covalently interacts with NFκB subunits and prevents nuclear translocation [123]. This drug diminished FcεRI-mediated IL-6 and TNF production as well as migration towards antigen in RBL-2H3 cells [124]. Similarly, pyrrolidine dithiocarbamate inhibits NFκB DNA binding and was shown to reduce HMC-1 mast cell cytokine production elicited by polychlorinated biphenyl compounds [125]. Therefore, this remains an area of potential growth for treating mast cell-associated inflammatory disease.

#### 2.4.2. Targeting NFAT

FcεRI also activates the NFAT transcription factor proteins [126]. Like NFκB, NFAT requires translocation to the nucleus to exert its gene regulatory functions. NFAT signaling is triggered by calcium influx. Once calcium ions are available in the cytoplasm, calmodulin activates calcineurin, a serine/threonine phosphatase. Calcineurin then dephosphorylates serine residues on NFAT proteins, allowing nuclear translocation [127]. NFAT has a central role in inducing IL-13 gene expression [126]. IL-13 is a Th2-type cytokine that increases airway hypersensitivity, mucus production, serum IgE levels, and eosinophil infiltration [128]. As IL-13 is implicated in allergic disease, NFAT is believed to be a reasonable therapeutic target. 

Targeting calcineurin is a proven means of inhibiting NFAT activity that has shown promise in suppressing mast cell function since the 1990s. Tacrolimus (FK506), a calcineurin inhibitor, inhibits IgE-induced histamine and PGD2 release from human mast cells [129]. Similarly, the calcineurin inhibitor cyclosporine A was shown to inhibit IgE-induced TNF secretion from mouse mast cells [130]. Both tacrolimus and cyclosporine A also inhibit ionomycin-mediated cytokine production from human and mouse mast cells [131,132]. These drugs also suppress c-Kit-mediated proliferation and show efficacy in mouse models of mast cell function, including allergic inflammation and parasite infection [130,133,134]. A more recent study found that tacrolimus was 100-fold more effective at suppressing IgE-mediated human mast cell function than cyclosporine but questioned whether the drug’s activity is due to calcineurin antagonism [135]. With the long safety record of these drugs and their apparent activity on mast cells, further study of calcineurin antagonism in allergic disease is warranted. This approach is now an alternative means of treating allergic eye disease and atopic dermatitis [136,137].

#### 2.4.3. Inhibiting the JAK-STAT Pathway 

Activated JAK kinases phosphorylate STAT transcription factor proteins, eliciting STAT translocation to the nucleus. While possessing myriad functions, STATs generally promote cell survival, proliferation, and pro-inflammatory gene transcription, including in mast cells [138]. Of the seven STAT proteins, STAT3, STAT5, and STAT6 have significant roles in mast cell biology. STAT3 is required for IgE-mediated degranulation in mouse and human mast cells, possibly through a role in the mitochondria [139,140]. STAT5, particularly, the STAT5B isoform, promotes mouse mast cell proliferation and survival, neoplasia, and IgE-mediated degranulation and cytokine secretion release [141,142,143,144]. This pro-stimulatory activity has linked STAT5 activation to atopic dermatitis and other atopic symptoms [145,146]. STAT6 promotes mast cell survival and IL-4 production [147,148]. With STATs the focus of drug development for inflammatory disease and cancer, new opportunities should be available for targeting mast cells. 

Inhibiting STAT phosphorylation blocks nuclear translocation. This can be accomplished by antagonizing JAK or other kinases that act on STATs or by directly inhibiting the STAT protein. Pimozide, a neuroleptic treatment, has off-target effects that prevent STAT5 phosphorylation without antagonizing STAT1 or STAT3 [149]. This drug inhibits IgE-induced rodent mast cell degranulation and migration in vitro, and cutaneous or systemic anaphylaxis in vivo [150,151]. A STAT3 inhibitor, C188-9 (also known as TTI-101), also reduced anaphylaxis in vivo, but its actions could be at least partly attributed to inhibiting histamine or platelet activating factor receptor actions on the vasculature [152]. Thus, while genetic evidence of a role for STAT3 is clear, drug-based treatment is less supported at this time. Finally, STAT6 inhibitors are being developed and are logical candidates for allergic disease because of the important role STAT6 plays in IL-4 and IL-13 signaling. While these have shown promise in mouse models of Th2 responses including atopic dermatitis and asthma, studies targeting mast cell function directly have not been published [153,154]. 

Distinct from the STAT pathway, JAK kinase inhibition also has great potential. Ruxolitinib, a JAK1/2 inhibitor, inhibited mast cell degranulation coincident with reducing food allergy symptoms in a mouse model [155]. This drug also improved quality-of-life measures for a systemic mastocytosis patient [156]. Tofacitinib (selective for JAK3) decreased IgE-mediated degranulation and cytokine production in the RBL-2H3 mast cell line and suppressed allergic conjunctivitis [157]. Similarly, oclacitinib (a JAK1 inhibitor) inhibited cytokine secretion and proliferation of canine mast cell tumor lines [158]. Finally, oclacitinib, tofacitinib, and momelotinib (a JAK1/2 inhibitor) all improved allergic dermatitis in a mouse model, including decreasing skin mast cell numbers [159,160]. These drugs hold tremendous promise and are reaching approval for use in canines. There is optimism to be found in clinical trials, as two small (n = 6 and n = 69) studies of atopic dermatitis patients showed significant improvement in clinical scoring that lasted up to 29 weeks [161,162]. Three JAK inhibitors—ruxolitinib, upadacitnib (JAK1 inhibitor), and abrocitinib (JAK1 inhibitor)—recently gained FDA approval for atopic dermatitis treatment in the U.S [163]. Further use in allergic disease appears to be likely in the near future. 

#### 2.4.4. Estrogen Receptor Signaling

Some atopic diseases such as allergic asthma exhibit female-biased incidence and severity [164]. This prompts questions about a role for sex hormones. Mast cells interact with 17β-estradiol (E2), the main circulating estrogen species. Once stimulated by E2, the ERα and ERβ receptors function as transcription factors that regulate inflammatory genes [165]. A correlation of E2 activity and mast cell function was made while investigating ovarian endometrioma [166]. E2 exposure increased FcεRI-mediated beta-hexaminidase and leukotriene C_4_ (LTC_4_) secretion in RBL-2H3 and HMC-1 cells. These proinflammatory mediators are responsible for bronchoconstriction during allergic diseases [167]. In addition to the previously mentioned mediators, the mRNA levels of stem cell factor (SCF), transforming growth factor β (TGF-β), and the chemoattractant MCP-1 are elevated by E2 treatment of RBL-2H3 cells [166]. Together these mediators promote mast cell recruitment and airway remodeling, so investigating potential methods of interference could lead to promising allergic disease therapeutics. 

Estrogen antagonists have been thoroughly investigated for the treatment of breast cancer, but their efficacy in treating allergic inflammation is still under investigation. The pioneer drug in this class, tamoxifen, antagonizes E2 interactions with its receptors but also has unrelated inhibitory effects on chloride ion channels [168]. Calcium and chloride channels are critical regulators of mast cell proliferation and mediator release [166,169]. Tamoxifen inhibited the outward flow of chloride by a CIC-5-like channel on HMC-1 cells and human lung mast cells, which corresponded to decreased proliferation [170]. Another study found that tamoxifen could reduce rat mast cell degranulation elicited by compound 48/80, substance P, or ionomycin [171]. In sum, there much to learn about estrogen signaling in mast cells and enough preliminary data to suggest it enhances inflammatory functions. With known sexual dimorphisms in allergic disease, this is an area that warrants consideration.

#### 2.4.5. The Complex Effects of PPAR-γ Function

Peroxisome proliferator-activated receptor-γ (PPAR-γ) is a ligand-activated transcription factor first identified for its role in lipid metabolism. PPAR-γ ligands include prostaglandins, fatty acids, and environmental pollutants [172]. PPAR-γ also has complicated roles in the immune system. It inhibits mast cell FcεRI responses, suppresses mucus production, and increases Treg numbers in adipose tissue [172]. These anti-inflammatory functions are partly mediated by inhibiting NFκB translocation and function. Therefore, drugs that activate PPAR-γ could be a promising development for IgE-mediated allergic disease incidences. However, as reviewed recently, PPAR-γ enhances Th2 responses, partly by increasing ST2 expression on Th2 and ILC2 cells [172]. These conflicting roles may be linked to differences in immune responses within and outside of adipose tissue. Clearly there is much left to unravel about this enigmatic protein.

Thiazolidinediones, a class of PPAR-γ agonists, suppress mouse mast cell maturation and induce apoptosis [173]. Pioglitazone, a synthetic PPAR-γ agonist, stabilized mice body temperature in an IgE-mediated passive systemic anaphylaxis model [174]. Pioglitazone demonstrated similar anti-inflammatory effects in a cockroach allergen-induced asthma model, suppressing airway hyperresponsiveness, serum IgE, and IL-4, IL-5, TNF-α, and TGF-β levels [175]. Another thiazolidinedione, Rosiglitazone, inhibited mast cell degranulation, reduced IL-4, IL-5, and TNF-α secretion, and IgE and IgG_1_ serum levels, and suppressed NFκB p65 subunit expression in a food allergy model [176]. Considering additional thiazolidinediones for allergy therapy may prove effective since PPAR-γ has multiple ligands and direct links to mast cell signaling. 

### 2.5. Dopamine Signaling

As residents of the central and peripheral nervous systems, mast cells interact with neurons and neuronal support cells and respond to neurotransmitters. Mast cells express the dopamine receptors D1 (DR1) and D2 (DR2) and can regulate neuroinflammatory diseases such as Alzheimer’s and Parkinson’s disease [177]. In a murine model of atopic dermatitis, signaling via the DR1 or DR2 receptor promoted mast cell degranulation and Th2 cell differentiation [178]. Dopamine signaling can induce IL-4 mRNA and reactive oxygenated species (ROS) generation, therefore antagonizing dopamine receptors on mast cells may attenuate inflammation [178]. 

Perphenazine, an anti-psychotic therapy, is a dopamine receptor antagonist currently being used for murine Th2-type allergic dermatitis. This FDA-approved antagonist successfully reduced serum IgE, histamine, and IL-4, IL-5, IL-13 transcription, and secretion in a 12-o-tetradecanoylphorbol-13-acetate (TPA) and oxazolone (OXA)-induced dermatitis mouse model. Perphenazine also inhibited mast cell migration to lesion sites [179]. Another DR1 and D2R antagonist, berberine, is being utilized for its anti-inflammatory and antidepressant effects. Treatment with berberine attenuated 2,4-dinitrofluorobenzene (DNFB)-induced allergic contact dermatitis in rats and significantly reduced ear swelling, mast cell recruitment, serum IgE, and p38 phosphorylation [180]. Berberine is popular in Chinese traditional medicine as a gastrointestinal remedy and has potential benefits for allergic diseases [181]. 

### 2.6. Statins and the Isoprenyl Lipid Pathway

Statins are one of the most prescribed drugs in America and the leading therapy to treat hypercholesterolemia and reduce cardiovascular morbidity and mortality [182]. The first statin to be described, Mevastatin, was derived from *Penicillium citrinium* extract in the 1970s by Akira Endo [183,184]. Since then, three other statins (pravastatin, simvastatin, and lovastatin) have been derived from the fermentation of fungi and five other statins have been synthesized chemically (atorvastatin, vervastatin, pitavastain, fluvastatin, and rosuvastatin) [185,186,187]. Statins inhibit the rate limiting step of cholesterol synthesis, 3-hydroxymethyl-3-methylglutaryl coenzyme A reductase (HMGCR) [188] (Figure 1). By slowing cholesterol synthesis, these drugs effectively reduce the circulating levels of low-density lipoprotein cholesterol, actions tied to reduced cardiovascular morbidity and mortality [189]. Over the last decade, however, statin efficacy has been also attributed to their ability to reduce inflammation [190,191]. One meta-analysis study found that statin use among patients who later suffered sepsis and other infections correlated with a protective outcome [192]. Another study found that increasing both dose and duration of statin use reduced the hazard ratio for emergency department visits and hospitalizations attributed to asthma [193]. Finally, one study using a mouse model of allergic airway inflammation and airway hyperreactivity showed that simvastatin reduced the inflammatory phenotype [194]. These studies and other meta-analyses of patient data suggest a role for statins in reducing inflammation, possibly including allergic disease. 

Many studies have linked statin effects to immune modulation [195,196]. Link et al. found that 20 mg/day of rosuvastatin therapy reduced Th1-related TNF and IFNγ levels [197]. Other studies have found that statins can reduce dendritic cell, T cell, mast cell, and macrophage activation in vivo and in vitro, independent of reduced circulating LDL levels [198,199,200,201,202]. Using human and mouse mast cells, we and others found that fluvastatin reduced IgE-mediated degranulation and cytokine production in vitro and suppress passive systemic anaphylaxis in mice [198,203]. 

Our work found that statin effects were not phenocopied by inhibiting cholesterol synthesis alone and that genetic background could greatly alter statin responsiveness, possibly owing to HMGCR upregulation [198]. Other studies have reported a similar finding, linking reduced statin efficacy to a gene variant that is regulated by cholesterol concentration, with reduced cholesterol eliciting increased HMGCR [204,205,206]. It is important to understand the mechanism of action because the literature is equivocal on statin efficacy in allergic diseases like asthma. While three large studies (n > 10,000) found significant improvements in asthma outcomes associated with statin therapy, smaller studies have yielded mixed, if generally positive, results [193,207,208,209,210,211,212,213]. One explanation is varied efficacy among statin drugs, since most studies analyze groups of patients taking any statin. In fact, we noted varied statin effects on mouse mast cells, with some statins having no effect on IgE-mediated activation while others were potent inhibitors [198]. This variability, coupled with genetically based resistance suggests that effective statin use for allergic disease requires a detailed understanding of drug mechanism. 

One important intermediate of cholesterol biosynthesis that could explain statin effects on inflammation is protein isoprenylation, including the Ras superfamily [214] (Figure 1). Isoprenylation is the addition of a 15-carbon chain by the enzyme farnesyl transferase (FT) or 20-carbon chain by geranylgeranyl transferase (GGT-1). This covalent modification occurs on the C-terminus of proteins bearing a CAAX motif, where C is cysteine, A is any aliphatic amino acid and X is any amino acid [215]. FT prefers a methionine in the final position, while GGT-1 prefers a leucine. A third transferase protein, geranylgeranyl transferase type-2 (GGT-2) can add two 20 carbon chains to proteins ending in a CCAX motif to Rab family proteins [216]. 

Isoprenylation is crucial for over 400 proteins to associate with membrane phospholipid bilayers [217,218]. When treating plasmacytoid dendritic cells with statins, it was found that they produced significantly less type 1 IFN when stimulated with TSLP, an important step in a Th2 allergic response. Statin treatment reduced STAT6 and NFκB phosphorylation and could be mimicked by inhibitors of geranylgeranyl transferase or Rho, suggesting a role for Rho geranylgeranylation in DC function [219]. Isoprenylation has also been linked to airway remodeling in allergic disease. Takeda et al. found that treating with clinically relevant simvastatin concentrations inhibited human airway smooth muscle cell proliferation through a GGT- and Rho-dependent mechanism [220]. We found that fluvastatin’s inhibitory effects on IgE-mediated mouse mast cell function could be phenocopied by inhibiting GGT-1 [198]. These studies, along with others in T cells, show an important role for isoprenylation, a product of the cholesterol synthesis pathway inhibited by statins, in inflammation and allergic disease [221]. 

Identifying the critical isoprenylation targets regulating mast cell activation will yield fundamental insight into mast cell biology. Among the >400 isoprenylation targets are Rab, Ras, and Rho family proteins [222]. In particular, Ras proteins have been linked to mast cell activation, while Rho proteins have been linked to mast cell cytoskeleton remodeling [223,224]. Ras involvement in IgE signaling has been demonstrated in mouse and human mast cells [225,226]. Specifically, K-Ras has been shown to play a significant role in mast cell survival and activation via c-KIT signaling [227]. Transcriptomic data show mouse mast cells express mRNA for K-Ras, N-Ras, and H-Ras [228]. Once Ras is activated, it signals through the classical Ras/Raf-1/MEK/ERK cascade [229,230]. Among Rho family candidates, Rac2 is a key player in cytoskeleton remodeling crucial for degranulation and histamine H4 receptor-mediated chemotaxis [221,231,232]. Another Rho family protein of note is Cdc42, which also plays a role in degranulation and chemotaxis [233]. Rab family proteins, which are isoprenylated through GGT-2, include Rab3d and Rab5, which have important roles in IgE-mediated mast cell granule formation and release in allergic disease [234,235]. Determining which of these proteins is responsible for the inhibitory effects statins could yield new clinical targets.

If statin effects are due to loss of isoprenylation, targeting this process directly may prove fruitful. There are inhibitors in all stages of development, including a few that are FDA-approved, for inhibiting isoprenylation [236,237]. There are four classes of isoprenylation inhibitors: GGT-2 inhibitors, GGT-1 inhibitors (GGTIs), FT inhibitors (FTIs), and dual farnesyl and geranylgeranyl transferase inhibitors (FGTIs) targeting both GGT-1 and FT. These classes have been highly studied and developed for possible cancer therapies, with the newest iterations of these drugs highly selective to their targets [217,222]. These inhibitors offer new possibilities for isoprenylation inhibition and therapeutic options. 

Overall, these studies show the ability of statins to reduce inflammation in allergic disease and suggest the effects are due to reduced isoprenylation of key proteins crucial for cell signaling. These findings and the known roles of isoprenylated proteins indicate the need for increased study of isoprenylation in allergic disease and the possible therapeutic relevance of isoprenylation inhibitors. 

### 2.7. Antidepressants as Mast Cell Inhibitors in Allergic Disease

Depression affects millions of people around the world and can be characterized by changes in sleep, severe weight fluctuation, and feelings of worthlessness [238]. In a study done by the CDC from 2015–2018, 13% of American adults reported taking an antidepressant in the last month; a significant increase from previous years [239]. Antidepressants are the most prescribed medication for depression and include tricyclic antidepressants (TCAs), monoamine oxidase inhibitors (MAOIs), serotonin-norepinephrine reuptake inhibitors (SNRIs), and selective serotonin reuptake inhibitors (SSRIs) [238]. SSRIs are the most widely used antidepressants, likely due to their increased safety and efficacy compared to the other classes [238]. Since 10% of Americans are taking an antidepressant, it could be beneficial to study other positive effects of these drugs. 

Importantly, there are consistent if still unclear links between depression and the immune system. Depression is increased in immune disorders, and reciprocally, IL-6 and C-reactive protein are upregulated in depressed individuals [240,241,242]. SSRIs are also able to decrease nitric oxide and TNF production by mouse microglia during IFNɣ stimulation [243]. Further, meta-analysis data show that anti-inflammatory treatment can help alleviate depression, and antidepressants improve COVID-19 symptoms [244,245]. In addition, there are data that show elevated inflammation in individuals who respond poorly to SSRIs [246]. In relation to mast cell function, MAOIs can suppress mouse mast cell degranulation, and SSRIs can decrease IL-1β and TNF mRNA expression [247,248]. These links led us to determine if antidepressants suppress mast cell-dependent allergic inflammation.

Fluoxetine is a commonly prescribed SSRI used to treat depression since 1988 [249]. It can also be used to treat obsessive compulsive disorder (OCD) and bulimia nervosa [249]. In addition to treating psychological disorders, there are data showing fluoxetine can suppress immune responses [250,251]. In a study done by Sherkawy et al., fluoxetine decreased airway inflammation in vivo [250]. Rats were sensitized with ovalbumin intraperitoneally before intranasal challenge along with 20 mg/kg fluoxetine treatment. They found that fluoxetine-treated rats had decreased serum IgE, IL-4 and TNF levels, and less airway obstruction noted in lung histology [250]. These data indicate that antidepressant treatment can suppress airway inflammation in vivo. Citalopram, another SSRI, also has anti-inflammatory effects [251]. Citalopram was FDA approved in 1998, and like fluoxetine, can also be used to treat OCD and some anxiety disorders [252]. Using a long-term treatment model, mice were given 10 mg/kg of fluoxetine or citalopram daily for four weeks intraperitoneally (i.p.) before splenocyte isolation [251]. Upon stimulation, citalopram-treated splenocytes produced significantly more anti-inflammatory IL-10 than controls [251]. 

These data suggest inflammation can be suppressed by SSRIs. Clinical evidence is supportive if not robust. One small study showed that among 21 patients with severe asthma, 12 weeks of escitalopram treatment improved asthma control and decreased inhaled corticosteroid use [253]. Another study of 42 severe asthmatics showed 60% had improved asthma symptoms after 6 weeks of fluvoxamine addition to their standard care. This study lacked a control group, however [254]. It is also worth noting that among a large (>60,000) cohort of asthma patients in Europe, severe asthmatics were twice as likely as controls to be using an antidepressant, suggesting these drugs are safe for use in this population [255]. 

In addition to SSRIs, other antidepressants have shown promise in treating allergic inflammation. In a study done by Zhang et al., allergic rhinitis was studied by sensitizing BALB/c mice with ovalbumin for three weeks, then challenging for five weeks with or without the TCA desipramine [256]. Desipramine-treated mice not only had diminished nasal symptoms such as sneezing and scratching, but also had reduced cytokines, decreased Ova-specific IgE levels, and less eosinophil influx [256]. Histology from the nasal mucosa revealed less inflammation in desipramine-treated mice, and flow cytometry found fewer Th17 cells [256]. The SNRI venlafaxine was shown to suppress humoral and cell-mediated immunity as well as pro-inflammatory cytokine levels in the experimental autoimmune encephalomyelitis (EAE) model of multiple sclerosis (MS) [257,258]. Mast cells are an important player in MS by mediating inflammation [259]. Mice receiving 60 mg/kg oral venlafaxine showed less brain inflammation, reduced TNF and IFNɣ, and lower EAE scores than untreated mice [258]. 

Overall, this research indicates that antidepressants of different classes (SSRIs, SNRIs, and TCAs) can suppress inflammation in models known to involve mast cell activation. Therefore, we support further studies into the viability of repurposing antidepressants for treating allergic disease.

### 2.8. MRGPR Proteins Driving Mast Cell Activation

This review has adhered to allergy-associated mast cell function. It is worth noting that mast cells express multiple receptors that largely function outside of allergic disease and generally promote protective functions. These include Fcγ receptors, Toll-like receptors, CD48, CD300a, and many G-protein coupled receptors. These have been discussed in recent reviews [260,261,262]. Among the non-IgE-mediated receptors that activate mast cells is one generating considerable interest and review articles, include a recent one in this journal [263]. MRGPRX2 was identified in 2006 as a receptor causing IgE-independent mast cell activation [264]. Since then, dozens of papers have mapped the pro-inflammatory role of MRGPRX2 and its mouse ortholog, MRGPRB2, which was identified in 2015 [265]. MRGPR-mediated signals elicit what are often termed pseudo-allergic reactions because they mimic IgE-mediated responses including urticaria, itch, and edema but lack the adaptive immune response that defines atopy. Thus, MRGPRX2/B2 do not formally elicit allergy, but manifest clinical symptoms that overlap considerably. For this reason, the signaling cascade has prompted much interest.

There are several aspects of MRGPRX2/B2 biology that make for a challenging study subject—and one that has so far escaped the development of effective inhibitors. First, MRGPRX2 and B2 have a growing list of >20 ligands that includes peptidermic and small molecule drugs, Substance P, Compound 48/80, bactericidal peptides such as cathelicidin, and several small, charged peptides derived from protein degradation [266]. These ligands likely have varying binding sites on the receptor, making complete inhibition with a single small molecule problematic. Second, MRGPR receptors appear to be selectively expressed on connective tissue mast cells (CTMC) in the mouse and chymase-tryptase-positive mast cells (MCTC) in humans. This presents technical challenges because the common mouse model of bone marrow-derived mast cells (BMMC or BMCMC) has only come CTMC characteristics and yields a relatively weak response to MRGPRB2 ligands [267]. Perhaps more importantly, mouse MRGPRB2 has only 53% sequence similarity with human MRGPRX2 and is part of a 22-member mouse gene cluster related to the human 4-gene cluster [264,268]. For these reasons, mouse studies must be interpreted with caution, as MRGPRB2 biology likely differs from its human counterpart.

As recently reviewed by Ogasawa and Noguchi, there is considerable interest in developing MRGPRX2 inhibitors [266]. In some regards, this should be feasible because the receptor has relatively low affinity for many ligands. A dozen MRGPRX2 inhibitors have been reported and include naturally occurring substances such as paeoniflorin and quercetin, the tripeptide QWF, DNA-based aptamers, and the heterocyclic “Compounds 1 and 2” identified by a library screen [269,270,271,272,273]. Most of the current inhibitors have off-target effects or remain to be tested in human models. Thus, it is too soon to state a clinically useful inhibitor exists, but there is reason for optimism. This is especially true if the clinical goal is to antagonize a small set of MRGPRX2 ligands such as peptidermic drugs that elicit anaphylaxis.

## 3. Discussion

Our purpose in this review is to summarize the most recent advances in allergy therapy and to suggest novel interventions based on drug repurposing. We have given particular focus to drugs disrupting mast cell function, since this has been an effective approach to treating allergy for decades. These are visually summarized in Figure 2. Allergic disease is a very common disorder in need of new treatments. Most estimates place allergic disease incidence at more than 50 M in the US alone, with an asthma incidence of >25 M and >4000 deaths in 2020 [274]. Food allergy has risen very quickly in the past generation, now affecting approximately 8% of US children and 11% of US adults [275,276]. While these data demonstrate the large number of patients affected, it is important to note several disparities. Using asthma as an example, CDC statistics from 2020 show the incidence among boys and girls was approximately equal in 2020, with 2.1 M in each sex, a rate of approximately 6% for each. However, the adult female rate (13.5 M, 10.4%) is nearly twice the adult male rate (7.5 M, 6.2%). Additionally, asthma rates are highest among the poor as well as Black or Native Americans [277]. As recently discussed by Warren et al., similar variations are also found in food allergy [277]. Such disparities emphasize the importance of finding interventions that can be made broadly available without major barriers to access. 

Humanized monoclonal antibodies represent a breakthrough in allergy therapy, with omalizumab being at the forefront since its first FDA approval in 2003. The ability to antagonize IgE interactions with its high affinity receptor allows intervention at a relatively early stage in allergic disease, preventing activation of mast cells and basophils, two major sources of allergic mediators. Preventing cell activation should logically result in broader effects than antagonizing individual mediators such as histamine or leukotrienes. Omalizumab also confirmed some basic biology, such as the critical role of IgE in surface FcεRI expression. In early trials, 3 months of omalizumab therapy reduced basophil FcεRI levels by 95%, an effect that has since been shown to occur on cutaneous mast cells and is sustained for long periods [278,279]. Ligelizumab is a logical improvement in this approach, as it appears to displace receptor-bound IgE and thus should be a faster and perhaps more effective therapy [35]. Dupilumab represents an even more fundamental attack on allergic disease, suppressing IL-4 and IL-13 signaling required for not only IgE production, but the underlying Th2 response driving it. These drugs are truly transformational and are improving the quality of life for patients. Their impact should only grow as they gain approval for more allergic disease states. Despite this, monoclonal antibodies are a difficult intervention to implement when health disparities are considered. For example, Buendia and colleagues recently discussed low-cost alternatives to Xolair, which has an annual cost of approximately USD 10,000 [280]. Dupilumab is significantly more expensive than Xolair. Broad advances in quality of care for the overall population will benefit more from inexpensive interventions. For these same reasons, it will be interesting to see if the potential to block IL-4, IL-13, or IgE functions via vaccinations or DARPin-based drugs continues to develop.

Drug repurposing (also known as repositioning) is an enticing consideration, especially for diseases like allergy for which much fundamental biology has been uncovered. As we know the critical cells and molecular targets, predicting which FDA-approved drugs may be effective is completely plausible. Additionally, these FDA-approved drugs have known safety profiles and side effects. However, caution is still warranted, since introducing these drugs into the inflammatory environment could have unforeseen consequences—especially for drugs not intended for anti-inflammatory purposes. Additionally, there is no current means to deliver most of the drugs discussed specifically to mast cells. This approach can dramatically improve the costs and time involved in drug testing and approval. As described by Nicola Nosengo, repurposing a drug can take 6–7 years and require several hundred million dollars [8]. This is half the time and perhaps 10% of the costs a new drug requires. Repurposing is a relatively new approach, with the first formal success being aspirin’s use as an anticoagulant in the 1980s (Reviewed in [281]). There are now dozens of successful repurposing examples affecting a wide range of diseases including cancer, erectile dysfunction, and obesity [282].

Among the possibilities for drug repurposing in allergic disease, our review of the literature suggests a few warranting strong consideration. The BTK inhibitor ibrutinib is an exciting possibility. This small molecule inhibitor targets a well-established pathway in mast cell and B cell biology, and has off-target effects on ITK, suppressing T cells. As discussed above, early studies in allergic patients are encouraging and the drug is relatively well tolerated. While its current cost is high, an oral drug would avoid some issues of access inherent with injectable monoclonal antibodies. Similarly, JAK inhibitor therapy is showing great promise and already approved for atopic dermatitis. This gives hope that this small molecule inhibitor, which inhibits mast cells, B cells, and T cells, may be expanded for use in more allergic diseases.

Our lab is most excited about the possibilities of statins as an addition to standard care for poorly controlled allergy and asthma. These drugs are widely prescribed, have an excellent safety profile, and have been shown in multiple large-scale studies to decrease key end points of asthma—emergency room visits, hospitalization, and oral steroid use. We found wide variations in responsiveness among mast cells from different mouse strains, human donors, and among various statins tested [198]. For this reason and because statin resistance is well documented, we suspect that only some statins will prove effective, and genetic background may prove critical. Moreover, our data suggest that blocking isoprenylation, not cholesterol, explains statin benefits. Our longer-term hope is that the importance of isoprenylation in cancer will yield new selective inhibitors that can be repurposed for allergic disease. 

In closing, we encourage basic and clinical researchers to consider these new avenues for allergy therapy. We also emphasize the critical role basic science continues to play in improving patient care. Xolair and dupilumab were built on a detailed understanding of ligand-receptor interactions that likely seemed esoteric when first published. BTK and JAK kinases have been precisely positioned in cell signaling cascades by decades of meticulous work that now provides a foundation for broadening and repurposing their use. It is our hope that the dramatic rise in allergic disease in the past two generations can be addressed partly by applying a fundamental understanding of disease to repurposing drugs already at our disposal.

## Figures and Tables

**Figure 1 cells-11-03031-f001:**
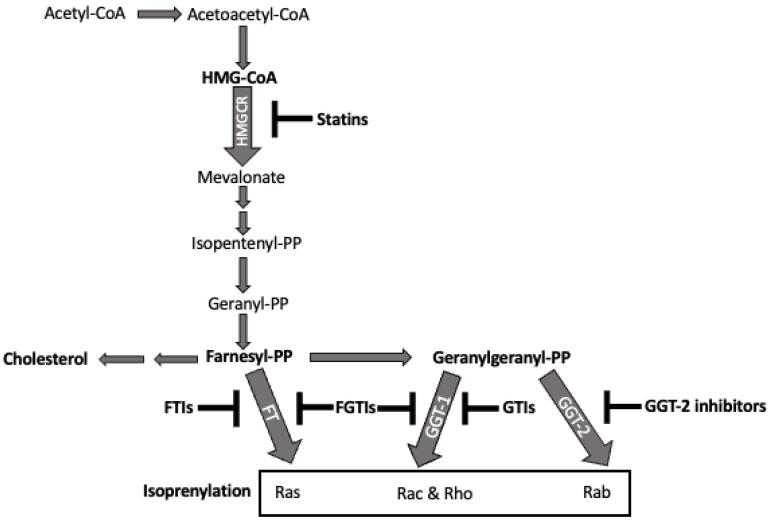
**Cholesterol synthesis and relevant inhibitors.** Figure depicts an abbreviated version of cholesterol synthesis with isoprenyl lipids in bold font. Statins act at the rate limiting step of mevalonate synthesis. Ras family proteins are substrates for the indicated isoprenyl transferases, which can be targeted by multiple drugs. FTIs = farnesyl transferase inhibitors; FGTIs = dual inhibitors of farnesyl transferase and geranylgeranyl transferase-1 (GGT-1); GTIs = GGT-1 inhibitors.

**Figure 2 cells-11-03031-f002:**
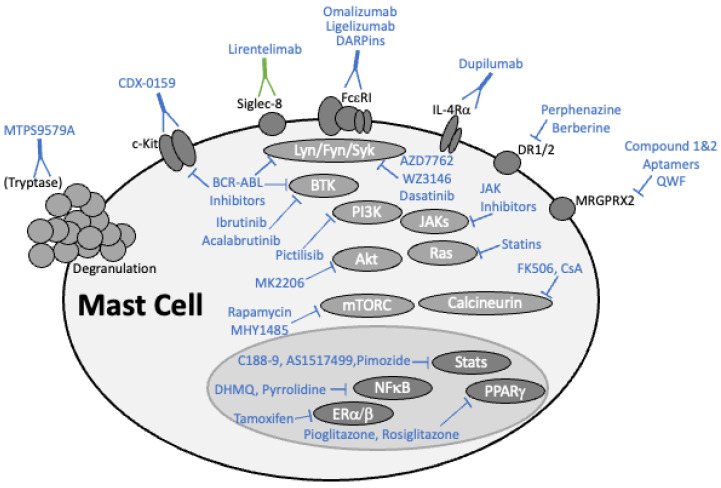
**Visual representation of drugs with known or potential ability to disrupt mast cell function.** Monoclonal antibody-based drugs are shown as Y-shaped symbols. As described in the text, many of the drugs are shown with their known or proposed off-target effects (e.g., BCR-Abl inhibitors with known effects on many kinases). All drugs shown have antagonistic effects except lirentelimab, which is depicted in green to indicate its agonist actions on the inhibitory receptor Siglec-8.

**Table 1 cells-11-03031-t001:** Monoclonal antibodies with known or likely uses in allergic disease.

Drug	Target	FDA Status
Omalizumab	IgE	Approved for some allergic disease
Ligelizumab	IgE	Phase 3 trials for allergic disease
Dupilumab	IL-4Ra	Approved for some allergic disease
Rituximab	CD20	Approved for some autoimmune diseases and B cell malignancies
Lirentelimab	Siglec-8	Phase 3 trials for allergic disease
MTPS9579A	Tryptase	Phase 2 trials for allergic disease
CDX-0159	c-Kit	Phase 1 and 2 trials for allergic disease

**Table 2 cells-11-03031-t002:** Kinase inhibitors with known or likely uses in allergic disease.

Drug	Target	FDA Status
Imatinib	BCR-ABL	Approved for Chronic Myeloid Leukemia (CML); Phase 2 clinical trial for Asthma
Nilotinib	BCR-ABL	Approved for CML
Dasatinib	BCR-ABL	Approved for CML and Philadelphia chromosome-positive acute lymphoblastic leukemia (ALL)
Bosutinib	BCR-ABL	Accelerated approval for CML and ALL
Ponatinib	BCR-ABL	Approved for CML and ALL
Ibrutinib	BTK	Multiple FDA approvals; Phase 2 clinical trial for Food Allergy
Acalabrutinib	BTK	Approved for Chronic lymphocytic leukemia (CLL) and Small lymphocytic lymphoma (SLL); Phase 2 clinical trials for Food Allergy
AZD7762	Lyn/Fyn/Syk	Phase I clinical trials for Solid Tumors
WZ3146	Lyn/Fyn/Syk	No US trials listed
Dasatinib	Lyn/Fyn/Syk	Approved for leukemia
Pictilisib	PI3K	Multiple clinical trials for cancer treatment
MK2206	Akt	Multiple clinical trials
Rapamycin	mTOR	Multiple FDA approvals
MHY1485	mTOR	No US trials listed

**Table 3 cells-11-03031-t003:** Compounds targeting transcription factors relevant to allergic disease. All compounds are antagonists except where stated.

Compound	Target	FDA Status
DHMQ	NFkB	No US trials listed
Pyrrolidine dithiocarbamate	NFkB	No US trials listed
Tacrolimus (FK506)	Calcineurin	Multiple FDA approvals
Cyclosporine A	Calcineurin	Multiple FDA approvals
Pimozide	STAT5 (off-target)	Approved for Tourette’s syndrome
C188-9 (TTI-101)	STAT3	Phase 1 for multiple cancers
AS1517499	STAT6	No US trials listed
Ruxolitinib	JAK1/2	Multiple FDA approvals
Tofacitinib	JAK3	Multiple FDA approvals
Oclacitinib	JAK1	Approved for use in canines
Momelotinib	JAK1/2	Phase 3 for myelofibrosis
Upadacitinib	JAK1	Multiple FDA approvals
Abrocitinib	JAK1	Approved for atopic dermatitis
Tamoxifen	Estrogen E2 receptor	Approved for breast cancer
Perphenazine	Dopamine D1 and D2 receptors	Approved for psychotic disorders and severe nausea
Berberine	Dopamine D1 and D2 receptors	Multiple stages of trials for metabolic disorders
Pioglitazone	PPAR-g (agonist)	Approved for Type 2 diabetes
Rosiglitazone	PPAR-g (agonist)	Approved for Type 2 diabetes

## Data Availability

Not applicable.

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
