# Peer review of "Targeting Mast Cells in Allergic Disease: Current Therapies and Drug Repurposing"

_cells, 2022, doi:10.3390/cells11193031_

Round 1

Reviewer 1 Report

This manuscript by Burchett et al is a exiting review in the field of mast cells biology. There is not much review exist on this topic. Manuscript nicely described all the literatures relevant to this topic. 

I have couple of minor suggestions.

Authors should include one more figure showing all the drugs and their target molecules that author mentioned in the manuscript. It will a pictorial view of the manuscript.

In some occasion epsilon and gamma was written as "e" and "g". This need to be corrected. 

Author Response

We added new Figure 2, which shows the targets of the drugs discussed. This was a good idea - many thanks! I had thought it would be too "busy" to be helpful, but really feel the figure makes the paper clearer.

We have (hopefully) corrected the Greek letters where needed.

Reviewer 2 Report

Allergic disease is a common disease. Mast cells play an important role in allergic diseases, especially allergic inflammation. In the past studies, the treatment of allergic diseases mainly used drugs to maintain the stability of mast cells and inhibit the secretion of mast cells. Although some treatment methods were effective for some patients, many patients still could not find a treatment strategy. Therefore, it is very necessary to make a summary of the current research progress of mast cells. In this review, the author summarizes the latest therapies and new drugs in clinical trials for mast cells, puts forward the possible ways of "new use of old drugs" in the treatment of allergic diseases, and shows an optimistic attitude towards the treatment of allergic diseases in the future.

In order to better understand the important role played by mast cells in the occurrence of allergic diseases, it is suggested to add the following contents to this review so that readers can better understand mast cells and allergic diseases after reading this review.

1. It is suggested to increase the content of the role of mast cells in allergic diseases, because this is the premise of the treatment strategy.

2. It is suggested to increase the knowledge of the types of receptors expressed on mast cells and the release of related substances after activation of different receptors.

3. Increase the research content and literature related to mast cells and IL-4, IL-13.

4. Please add the contents of recent research on MrgprB2.

Author Response

We have made the suggested changes, which are highlighted in the current manuscript version. This includes improving the description of mast cell functions in allergy/asthma, more discussion of IL-4/IL-13, and a new section on MRGPRX2/B2. We also added a brief section describing other inflammatory receptors expressed on mast cells, but did not go into depth about these receptors. Our reasoning is that these receptors are not directly related to allergy, though they may enhance mast cell-mediated inflammation. Given the length and depth of our manuscript (275 references), I felt that trying to cover these additional receptors in depth was unrealistic. I hope the reviewer understand our perspective.

Reviewer 3 Report

General Comments:

In their review, Burchett and colleagues provide a broad overview of drug candidates that could be repurposed for the treatment of atopic diseases via targeting mast cells and/or mast cell activation.

The review is well written and comprehensive and fills a gap in the allergy/mast cell-literature by highlighting novel potential therapies.

I only have minor specific comments and suggestions (please see below).

In general, I think it would be a useful to include information on species (human or mouse/rat) and cell model (primary or cell line) when discussing in vitro mast cell data (this is already done in some cases).

While the suggestion to re-purpose drugs to target mast cells is certainly innovative and bears great potential, there’s always the danger of off target effects and side effects (which is logical since the drugs were not originally developed to target mast cells). In my opinion, this should be more emphasized in the review; it could be useful to have respective paragraphs in all sections starting with 2.2. Along this line, it might be interesting to propose or speculate how such broadly-acting drugs could be specifically delivered to mast cells (if such ways exist).

While the section regarding mTOR taretting (2.2.4) goes already in this direction, it might also be worth to mention (maybe in the discussion?) that metabolism-targeting drugs could be interesting candidates.

For tables listing the different drug candidates, it might be useful to state the trial IDs if relevant.

Specific comments:

·        *While the review already covers a number of important drugs and concepts that target type 2 immune responses, IgEs or mast cells, I would suggest to also include a (brief) section on DARPins (e.g. reported in PMID 33991582) to introduce this novel and promising concept to the readers.

·        Furthermore, I suggest to include the recent work of the Reber lab on anti IL-4 and IL-13 vaccines (i.e. PMIDs 33976140) for allergy treatment in the section on IL-4 and IL-13 targetting (2.1.2).

·         Figure 1: please include a list of compound group abbreviations used in the figure (FTI, FGTIs, etc)

·        Line 87: please shortly explain the function of FceRIa (IgE binding part) and that it’s part of the FceRI (and not a receptor itself).

·        Line 139: change “impoved in symptom” to “improved symptom”

·        Section 2.2.2: please mention whether ibrutinib leads to mast cell depletion (since FceRI signaling is also a survival signal) or “only” blocks IgE-mediated activation

·        Section 2.3.5: in several instances PPAR-g should be changed to PPAR-gamma (symbol); e.g. line 481

·        Line 523: change “describe” to “described”

Author Response

Reviewer 3 had many important suggestions that improved this manuscript. Changes are highlighted.

We clarified many sections in which the species of mast cells was unclear.

The risk of drug repurposing, specifically off-target effects, is now mentioned in the Discussion. Importantly, these FDA-approved drugs have known safety profiles and side effects, which is a significant benefit over novel compounds.

We added a little more discussion of metabolism targeting.

We chose to exclude clinical trial numbers because many of the drugs shown here have multiple trials, making the Table unwieldy. 

Thank goodness the reviewer caught my omission of DARPins and IL-4/IL-13 vaccination! These have been added.

Figure 1 now has abbreviations explained.

The FcERI alpha chain is now described as part of the tetrameric complex.

Ibrutinib's weak apoptotic effects are explained.

Grammatical issues (including Greek lettering) have been corrected.

Sincerely, this paper is better because of the careful reading Reviewer 3 did. Many thanks.

John Ryan